# An Innovative Approach to Determine Coastal Scenic Beauty and Sensitivity in a Scenario of Increasing Human Pressure and Natural Impacts due to Climate Change

Alexis Mooser [1,2], Giorgio Anfuso [2,*], Allan T. Williams [3], Rosa Molina [2] and Pietro P. C. Aucelli [1]

1   Department of Science and Technology (DiST), Parthenope University, 80143 Naples, Italy;
    alex.moosr@gmail.com (A.M.); pietro.aucelli@uniparthenope.it (P.P.C.A.)
2   Faculty of Marine and Environmental Sciences, University of Cádiz, Polígono Río San Pedro s/n,
    11510 Puerto Real, Spain; r.molina.gil@gmail.com
3   Faculty of Architecture, Computing and Engineering, University of Wales: Trinity Saint David (Swansea),
    Mount Pleasant, Swansea SA1 6ED, UK; allanwilliams512@outlook.com
*   Correspondence: giorgio.anfuso@uca.es; Tel.: +34-956-016-167

**Abstract:** Coasts worldwide face a great variety of environmental impacts, as well as increased anthropogenic pressures due to urbanization and rapid population growth. Human activities menace ecosystem services and the economy of coastal countries, often based on "Sun, Sea and Sand" (3S) tourism. The five parameters of greatest importance (the "Big Five") for beach visitors are safety, facilities, water quality, no litter and scenery, and the characterization of the latter was recently carried out by means of a checklist of 26 natural and human parameters, parameter weighting matrices and fuzzy logic, according to the "Coastal Scenic Evaluation System" (CSES) methodology. In order to propose sound coastal management strategies, the main aim of this paper is to propose a method to determine the scenic sensitivity of (i) natural parameters to coastal natural processes in a Climate Change context and (ii) human parameters to visitors' pressure in a scenario of increasing tourism and coastal developments. Regarding natural parameters, the sensitivity of "Beach face" and "Dunes" parameters is determined according to an Erodibility Index with a Correction Factor, taking into account wave forcing characteristics, tidal range and trends at a local scale of Sea Level Rise and Storm Surge. This establishes a Sensitivity Index to natural processes. A site's scenic sensitivity to human pressure/activities was determined by considering the sensitivity of several human parameters of the CSES method according to beach typology and access difficulty together with the Protection Area Management Category to which a site belongs. A Human Impact Index is obtained, which is afterwards corrected by taking into account local trends of tourism pressure, establishing a Sensitivity Index to human pressure. Finally, a total Sensitivity Index considering both natural processes and human pressure is obtained, and sites divided into three sensitive groups. The results can be useful to limit and prevent environmental degradation linked to natural processes and tourism development, and also to suggest measures to improve the scenic value of investigated sites and their sustainable usage. The method was tested for 29 sites of great scenic quality along the Mediterranean coast of Andalusia, Spain.

**Keywords:** landscape; beach; dune; Erodibility index; tourism pressure; coastal management; sustainability; Andalusia

## 1. Introduction

Travel and tourism is one of the largest growth industries in the world [1,2] and, by 2030, international tourist arrivals worldwide are expected to reach 1.8 billon [3]. Tourism average contribution to GDP is ca. 10% and reaches 25% for small islands and developing countries and it is responsible for the employment of 1 out of 10 worldwide jobs [3]. The United States, Spain and France top the rankings in terms of tourism receipts

and number of international visitors, Spain being second in both rankings [2]. Within the European Union, arrivals to the five emerging economies—Bulgaria, Poland, Hungary, Romania and Croatia—grew somewhat faster, at an annual rate of 8%, while receipts grew 10%, reaching EUR 29 billion. Many Caribbean and Mediterranean countries have developed proactive growth policies along the coastal area [4], with Spain, France, Italy and Greece accounting for "the most significant flow of tourists . . . a Sun, Sea and Sand (3S) market" [5] (p. 58). Beaches are a major player in tourist market; in recent years, around one third of the global tourist arrivals have been registered in Mediterranean countries predominantly along the coast [2,3].

Economic activities imply a pressure on natural areas but, in the case of coastal zones, several specific environmental issues arise, such as the proliferation of engineered structures, intensive use of natural shores for recreation and tourism, beach pollution, and extraction of sand and gravel for construction purposes, all fulfil important ecological, societal and economical functions, e.g., [6–12]. At the same time, such pressure deeply affects the global economy of coastal countries. Indeed, one of the most important functions of coastal ecosystems is the protection of human assets against storm surges and salt water intrusion, absorption of land-based nutrients and pollutants drained by rivers to the sea, and the breeding and feeding of fish, crustaceans and birds. To replace these naturally fulfilled functions would cost far more than that which future generations of European citizens could afford [8,13–15].

Economic activities have also contributed to accelerated coastal erosion, which is a natural process that has always existed and throughout history has helped to shape coastlines. There is now evidence that the current scale of coastal erosion is far from the natural one and is now one of the main problems of coastlines around the world [13,16]. Many studies have found that over 70% of the shorelines around the world are retreating and this trend is increasing because of climate change-related processes [16]. Diverse investigations relate coastal retreat with different parameters, such as maritime climate, sediment transport, sea level rise, etc., but it is unclear to what extent these factors influence coastal erosion [17]. For example, the Intergovernmental Panel on Climate Change (IPCC) has predicted an increase in sea level at a much faster rate than that really experienced in the first part of this century. At many locations, human attempts to remedy the situation, e.g., by emplacing breakwaters, groins or revetments, has made the situation worse by increasing erosion in downdrift areas, according to the "Domino" effect [18,19], and damaging natural landscape and coastal ecosystems in unexpected and unpredicted ways [10,13,20]. Pranzini and Williams [10] argued that coastal retreat in European countries was taken into account only when the coastal tourism boom appeared as an economically promising alternative. Beach width had to be maintained/increased, thereby securing a comfortable leisure surface for increasing human densities on a beach under the "3S" market [5]. This market is based, according to numerous questionnaires concerning beachgoers preferences [21], on five parameters (the "Big Five") that are of the greatest importance to coastal visitors [21–23]: safety, facilities, water quality, no litter, and scenery. Therefore, the latter, e.g., coastal landscape, is a vital component for the "3S" market and drives the economy of many coastal countries.

The elaboration of a method to preserve and enhance coastal scenic beauty is the main focus of this paper. Therefore, the determination of landscape characteristics, protection, conservation and management is a paramount and mandatory issue to challenge within coastal areas promoted in landscape conventions [24–28]. Even the European Commission, through the project, EUROSION, launched in 2004, recognised the relevance of the sustainable development of coastal zones and conservation of dynamic habitats, especially on the remaining undeveloped coast, as important long-term goals for European coastal zones. This requires a respect for, and, in many cases, the restoration of, the natural functioning of the coastal system and its natural resilience to erosion and flooding. Therefore, the aim of this paper is to determine a methodology to assess the resilience of sites of great scenic beauty to, from one hand, erosion and flooding processes in a Climate Change scenario

and, from the other, the increasing human pressure in order to propose sound management strategies. The method was tested to assess the sensitivity of 29 sites (Figure 1) of great scenic relevance located in the Mediterranean coast of Andalusia (Spain), the scenic characteristics of which were described in detail by Mooser et al. [29].

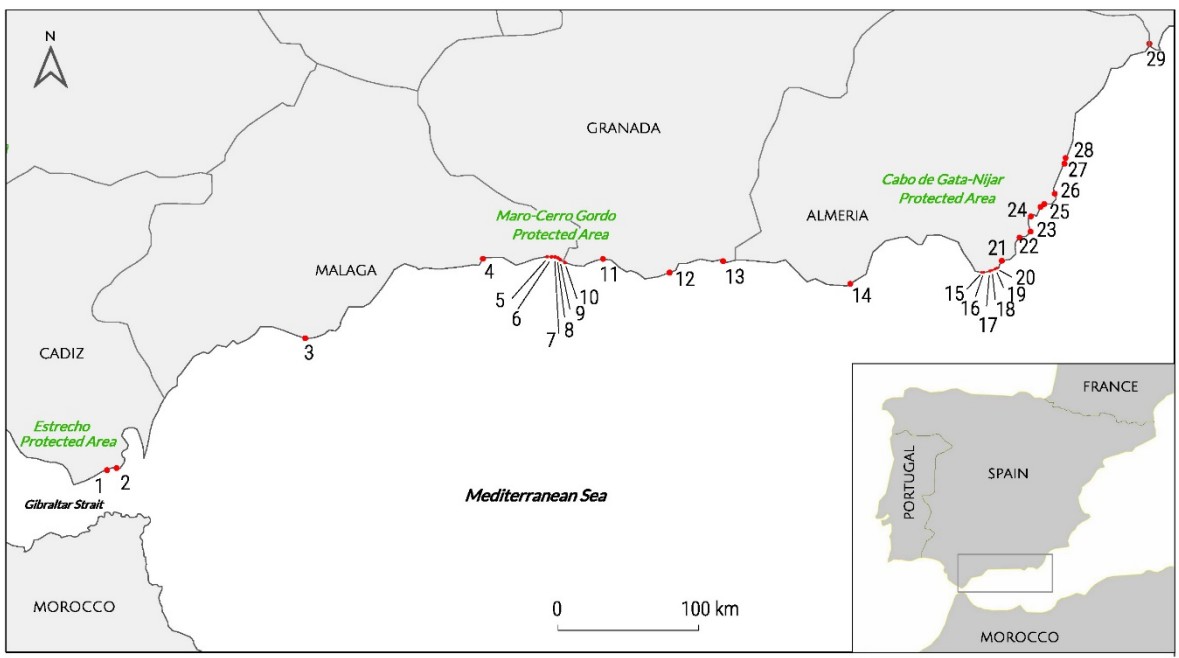

**Figure 1.** Location of the investigated sites along the Andalusia Mediterranean coast (Spain).

## 2. Study Area

The Andalusia Mediterranean coast, located in Southwest (SW) Spain, is ca. 597 km in length (350 km consist of sand sectors and 196 km of cliffs), and administratively includes the provinces of Cadiz, Malaga, Granada, and Almeria (Figure 1). It is a microtidal environment (<20 cm) exposed to winds and waves approaching from the southeast, and, secondarily, the SW [30,31].

The Betic Chain, a well-developed mountainous ridge that reaches high elevations very close to the coast (i.e., Sierra de la Plata in Gibraltar Strait Area, Sierra Tejeda, Almijara and Alhamada at Malaga and Granada provinces), strongly dominates coastal physiography. Several coastal plains are observed, especially at the mouth of short rivers and *ramblas* (seasonal stream) that drain the Chain. Beaches usually consist of fine and medium dark coloured sand and/or pebbles at *ramblas* mouths. In places, headlands and rocky sectors give rise to pocket beaches of different sizes, e.g., Maro-Cerro Gordo coastline (Figure 1 or Figure 2).

Two climatic zones stand out: (i) subtropical, along the coast of Cadiz, Malaga, and Granada; (ii) sub desert characteristics, at the province of Almeria. Regarding the former zone, the Betic Chain and coastal orientation favour average annual temperature of c. 13 and 19 °C in summer, with rainfall ranging from 400 to 900 mm/year—the most abundant values are observed at Gibraltar Strait. The latter zone presents extremely limited rainfall (ca. 200 mm/year) and average annual temperatures of 21 and 26 °C in July–August [32]. According to the Köppen classification, coastal areas of Cadiz, Malaga and Granada are categorised as Cfa (Humid subtropical climate), and Almeria as BSk (Cold semiarid climate). In Andalusia, weather conditions make the coastal environment very attractive to national and international tourism during several months per year.

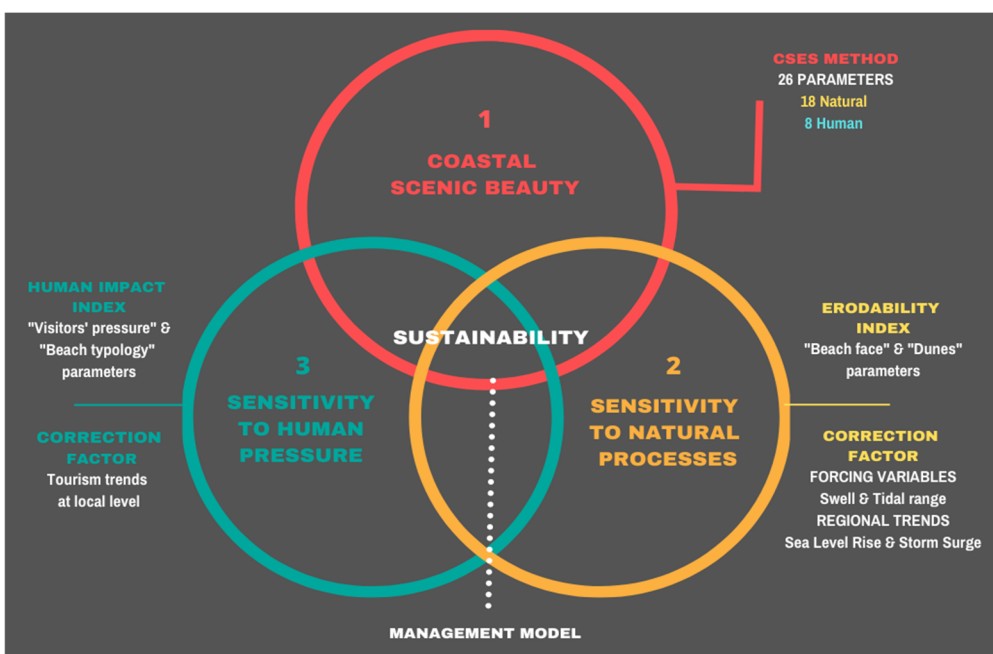

**Figure 2.** Summary of concepts and methods used in this paper.

Protected Areas are managed by the RENPA (Andalusia Network of Protected Areas) which deals with all of the issues related to the establishment and management of international, national, Natura 2000 and regional categories of protection located in Andalusia. The best known coastal protected areas are the Natural Parks of Estrecho de Gibraltar (in Cadiz province), Cabo de Gata-Nijar (Almería), and the Natural Place of Maro-Cerro Gordo in Malaga. Historical heritage along the coast is very important too. Several castles (e.g., at Cala San Pedro in Almeria) and *Vigia* towers, i.e., towers built in the XVI Century and used for coastal surveillance to prevent Berber pirate attacks (e.g., Cala Arena in Cadiz, Cala El Cañuelo in Malaga or Cala Rijana in Granada) can be observed along the studied coast. An exhaustive description of natural and human scenic characteristics can be found in Mooser et al. [29].

## 3. Methods

For this purpose, in a first step (Figure 2), coastal scenic beauty is proposed to be evaluated according to the well-known methodology "Coastal Scenic Evaluation System" (CSES) [21,33]. The CSES considers 18 physical and 8 human parameters that are classified from 1 (absence or poor scenic quality) to 5 (excellent/outstanding quality). This coastal scenic classification allows one to point out how scenic areas may be improved by judicious intervention relating to physical and anthropogenic parameters chosen for assessment. In a second step (Figure 2), the method aims to determine the sensitivity of sites—according to its own natural characteristics (e.g., beach width, presence of dunes, etc.)—to increasing coastal energy processes in a Climate Change context. A proposed third step (Figure 2), is a method to determine a site's sensitivity to human pressure/activities according to a worldwide increasing trend of beach visitors and the level of site protection (e.g., Natural Park, etc.). Information obtained in this study will constitute a basic requirement to improve knowledge of the landscape value of investigated sites, especially those belonging to natural places, bearing in mind that protected areas are one of the most attractive coastal tourist destinations that constitute not only an economic resource but also contribute to local people's quality of life [14]. Barbosa de Araújo and Da Costa [34] (p. 1440), working in Brazil, showed that, "landscape was probably highly rated as an attribute in visitor's choice," and scenic destinations for tourism purposes is now well ensconced in tourism literature, e.g., [35].

*3.1. Coastal Landscape Determination and Enhancement*

The Council of Europe [24] defines scenery as "the appearance of an area" and is a part of a coastal landscape inventory available for different coastal disciplines, such as, geography, geology, planning, etc. Likewise, coastal landscapes can be described as a littoral area, as perceived by humans, whose character results from the multiple interactions between natural and/or human factors [24]. A basic difficulty in assessing scenic quality is that of landscape definition, as it is a theoretical concept that is greatly confused by semantic difficulties, misunderstandings and controversies [36]. Steers [37] (p. 6) remarked that "any assessment of coastal quality is likely to meet with criticism", alas a comment still true in 2020. This is certainly a truism, as Teale [38] (p. 72) also argued that "nature affects our minds as light affects a photographic emulsion on a film. Some films are more sensitive than others, some minds are more receptive."

In this study, scenic assessment of most attractive sites was carried out according to the methodology "Coastal Scenic Evaluation System" (CSES), a semiquantitative analysis based on fuzzy logic analysis and parameter weighting matrices to eliminate subjectivity [21,39,40]. The technique is based upon a wealth of information collected from >1000 beach users chosen from random number tables in Malta, Turkey, and the UK [21,39], which gave rise to a large data base with a standard error of 0.03. As a result, we obtained a methodology based on a checklist of 26 natural and human parameters, parameter weighting matrices and fuzzy logic. Among the parameters, the CSES considers 18 physical components—cliff (height, slope, features), beach face (width, colour, type), rocky shore (slope, extent, roughness), dunes, valley, landform, tides, coastal landscape features, vistas, water colour and clarity, vegetation cover and debris—and eight human parameters—noise disturbance, litter, sewage evidence, built and non-built environment, access type and utilities (Table 1). All parameters were rated on a five-point attribute scale ranging from presence/absence or poor quality (1) to excellent/outstanding quality (5). Each parameter had a different weight, i.e., not all parameters are worth the same but the weight of all physical components is equal to human parameters. The weight components were identified from thousands of new questionnaires carried out in the UK and the Mediterranean, e.g., field testing showed that beach users prefer a white/golden sand beach than a dark one [33]. A Fuzzy Logic Approach (FLA) was adopted to overcome subjectivity and quantify uncertainty and each parameter has been associated with a membership-graded matrix to counteract potential errors in assigning grades, e.g., "Beach face" parameter (Figure 3).

**Table 1.** Checklist parameters (CSES) [21,39].

| No | Physical Parameters | | Weight | Rating | | | | |
|---|---|---|---|---|---|---|---|---|
| | | | | 1 | 2 | 3 | 4 | 5 |
| 1 | | Height (m) | 0.02 | Absent | $5 \leq H < 30$ | $30 \leq H < 60$ | $60 \leq H < 90$ | $H \geq 90$ |
| 2 | CLIFF | Slope | 0.02 | $<45°$ | $45–60°$ | $60–75°$ | $75–85°$ | circa vertical |
| 3 | | Features * | 0.03 | Absent | 1 | 2 | 3 | Many (>3) |
| 4 | | Type | 0.03 | Absent | Mud | Cobble/Boulder | Pebble/Gravel | Sand |
| 5 | BEACH FACE | Width (m) | 0.03 | Absent | $W < 5$ or $W > 100$ | $5 \leq W < 25$ | $25 \leq W < 50$ | $50 \leq W \leq 100$ |
| 6 | | Colour | 0.02 | Absent | Dark | Dark tan | Light tan/ bleached | White/gold |

**Table 1.** *Cont.*

| No | Physical Parameters | | Weight | Rating | | | | |
|----|----|----|----|----|----|----|----|----|
| | | | | **1** | **2** | **3** | **4** | **5** |
| 7 | ROCKY SHORE | Slope | 0.01 | Absent | <5° | 5–10° | 10–20° | 20–45° |
| 8 | | Extent | 0.01 | Absent | <5 m | 5–10 m | 10–20 m | >20 m |
| 9 | | Roughness | 0.02 | Absent | Distinctly jagged | Deeply pitted and/or irregular | Shallow pitted | Smooth |
| 10 | DUNES | | 0.04 | Absent | Remnants | Fore-dune | Secondary ridge | Several |
| 11 | VALLEY | | 0.08 | Absent | Dry valley | (<1 m) Stream | (1–4 m) Stream | River/ limestone gorge |
| 12 | SKYLINE LANDFORM | | 0.08 | Not visible | Flat | Undulating | Highly undulating | Mountainous |
| 13 | TIDES | | 0.04 | Macro (>4 m) | | Meso (2–4 m) | | Micro (<2 m) |
| 14 | COASTAL LANDSCAPE FEATURES ** | | 0.12 | None | 1 | 2 | 3 | >3 |
| 15 | VISTAS | | 0.09 | Open on one side | Open on two sides | | Open on three sides | Open on four sides |
| 16 | WATER COLOUR & CLARITY | | 0.14 | Muddy brown/grey | Milky blue/green | Green/grey/ blue | Clear/dark blue | Very clear turquoise |
| 17 | NATURAL VEGETATION COVER | | 0.12 | Bare (<10% vegetation) | Scrub/ garigue (marran, gorse) | Wetlands/ meadow | Coppices, maquis (±mature trees) | Varity of mature trees |
| 18 | VEGETATION DEBRIS | | 0.09 | Continuous (>50 cm high) | Full strand line | Single accumulation | Few scattered items | None |
| **Human Parameters** | | | | | | | | |
| 19 | NOISE DISTURBANCE | | 0.14 | Intolerable | Tolerable | | Little | None |
| 20 | LITTER | | 0.15 | Continuous accumulations | Full strand line | Single accumulation | Few scattered items | Virtually absent |
| 21 | SEWAGE DISCHARGE EVIDENCE | | 0.15 | Sewage evidence | | Same evidence (1–3 items) | | No evidence of sewage |
| 22 | NONBUILT ENVIRONMENT | | 0.06 | None | | Hedgerow/ terracing/ monoculture | | mixed cultivation ± trees/natural |
| 23 | BUILT ENVIRONMENT | | 0.14 | Heavy Industry | Heavy tourism and/or urban | Light tourism and/or urban | Sensitive tourism and/or urban | Historic and/or none |
| 24 | ACCESS TYPE | | 0.09 | No buffer zone/heavy traffic | No buffer zone/light traffic | | Parking lot visible from coastal area | Parking lot not visible from coastal area |

**Table 1.** *Cont.*

| No | Physical Parameters | Weight | Rating | | | | |
|---|---|---|---|---|---|---|---|
| | | | 1 | 2 | 3 | 4 | 5 |
| 25 | SKYLINE | 0.14 | Very unattractive | | Sensitively designed high/low | Very sensitively designed | Natural/historic features |
| 26 | UTILITIES *** | 0.14 | >3 | 3 | 2 | 1 | None |

\* Cliff Special Features: indentation, banding, folding, screes, irregular profile; ** Coastal Landscape Features: peninsulas, rock ridges, irregular headlands, arches, windows, caves, waterfalls, deltas, lagoons, islands, stacks, estuaries, reefs, fauna, embayment, tombola, etc.; *** Utilities: power lines, pipelines, street lamps, groins, seawalls, revetments, restaurants, etc.

$$M_s = \begin{bmatrix} 1 & 0 & 0 & 0 & 0 \\ 0 & 1 & 0 & 0 & 0 \\ 0 & 0.2 & 1 & 0.2 & 0 \\ 0 & 0 & 0.2 & 1 & 0.6 \\ 0 & 0 & 0 & 0.6 & 1 \end{bmatrix}$$

1: Stands for "Absent"
2: $W\langle 5\,m \; or \; W\rangle 100\,m$
3: $5\,m \le W < 25\,m$
4: $25\,m \le W < 50\,m$
5: $50\,m \le W \le 100\,m$

**Figure 3.** Grading matrices for "Beach face width" parameter (point 5, Table 1, CSES).

This enabled an Evaluation Index ("D") value to be calculated, establishing 5 scenic categories, ranging from Class I (excellent quality) to Class V (poor quality). The end points are −2 and +2, respectively. The higher the Evaluation index or "D" value, the more attractive the coastal scenery:

Class I. Extremely attractive natural sites with very high landscape values and $D \ge 0.85$.

Class II. Attractive natural sites with high landscape value and $0.65 \le D < 0.85$.

Class III. Mainly natural sites with few outstanding landscape features and $0.4 \le D < 0.65$.

Class IV. Mainly unattractive urban sites, with low landscape values and $0 \le D < 0.4$.

Class V. Very unattractive urban sites, with intensive development, a low landscape value and $D < 0$.

Assessment matrices were obtained for investigated sites and presented as histograms, membership degree and weighted average of attributes (Figure 4).

This method has been successfully used in Andalusia (Spain) by [29] and at many sites around the world [33,41–43]. A detailed description can be found in [33] that presented ca. 1000 sites around the world. Studies focused on characterizing and preserving the most attractive coastal scenic sites, have to be focused only to Class I and II sites. One of the main aims of the CSES method is to point out how scenic areas may be improved by judicious intervention relating to physical and anthropogenic parameters chosen for assessment. All improvements can be measured by the Evaluation Index ("D"). Most sites have physical parameters for which coastal zone managers can do little or nothing to alleviate scenic impact, e.g., presence of cliffs, vistas; a few exceptions being, for example, the formation in eroding coastal sectors of an artificial wide beach and dune ridges, as observed in Varadero (Cuba) by [41]. This fact favours the improvement of natural (points 4–6 and 10, Table 1) and human parameters (points 19 and 24), since dunes create a buffer zone between the beach and the urbanised environment. The rich geological heritage, often exposed to natural processes and man-made impacts, can ensure strong benefits and contributes to sustainable tourism development for local communities when well managed [44].

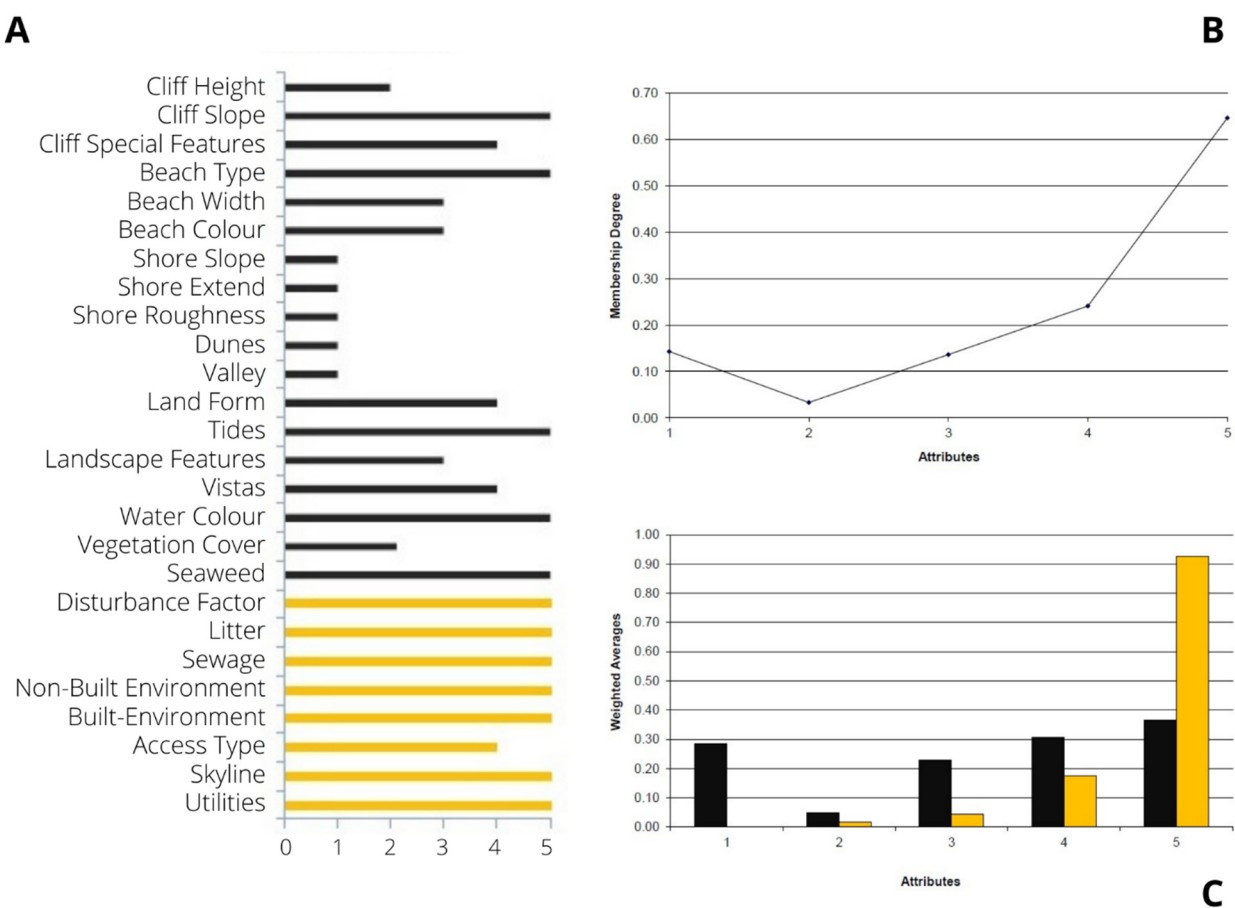

**Figure 4.** Scenic evaluation rating histograms for Cala Media Luna, Andalusia (Spain, Almería province; D: 1.01; Class I) (**A**), membership degree vs. attributes (**B**) and weighted averages vs. attributes (**C**).

Concerning human parameters, as an example, Rangel-Buitrago et al. [7] observed in Colombia that scenic characteristics can be improved by removing hard and disorganised existing protective structures that negatively affect point 26 (Table 1) together with reducing noise, beach litter and sewage discharge evidence (points 19, 20 and 21, Table 1). In many numerous previous studies, e.g., [29,41–43], and at many sites around the world, the CSES method has been used to efficiently characterise and improve coastal beauty and, within this paper, a further step forward is proposed to determine site sensitivity to marine agent impacts in a Climate Change context, together with human pressure.

### 3.2. Determination of Scenic Sites Sensitivity Processes

The aim of this paper is not to assess a coastal systems' protection function, i.e., their capacity of reducing the sensitivity/vulnerability of landward coastal ecosystems and/or human settlements (e.g., role/capacity of dune ridges in the protection of landward ecosystems or human activities/structures against flooding or erosion processes), but to determine the intrinsic sensitivity of coastal scenic parameters to erosion/flooding processes in a Climate Change context.

To this purpose, the present methodological approach comprises the following steps (Figure 5):

- In a first step, the sites investigated were divided into 3 categories according to their own physical characteristics (Figure 5).
- In a second step, an "Erodibility index" (EI) was proposed in order to calculate the level of sensitivity to natural processes of sites belonging to each category, Table 2.

- In a third step, a "Correction Factor" (CF) was applied to predict future relevance of energetic factors in a Climate Change scenario.
- In a fourth step (Figure 5), we finally established a "Sensitivity Index" (SI) of scenic sites to natural processes (Table 2).

**Table 2.** Equations regarding Erodibility Index (related to sites categories), Correction Factor and Sensitivity Index.

| Indexes and Categories | Equations | Parameters |
|---|---|---|
| Erodibility Index (1) for Category II sites ($EI_{C2}$) | $EI_{C2} = E_{BF} = \dfrac{\frac{Pn_1 + Pn_2 + \frac{Pn_{3a} + Pn_{3b}}{2}}{n_{Pn}} - 1}{A - 1}$ (1) | $E_{BF}$ erodibility of beach face parameters <br> $Pn$ : natural parameter <br> $Pn_1$ : dry beach evolution <br> $Pn_2$ sediment grain size <br> $Pn_{3a}$: rocky shore width <br> $Pn_{3b}$: rocky shore location <br> $n_{Pn}$: number of natural parameters (3) <br> $A$: maximum attribute value (5) |
| Erodibility Index (2) for Category III sites ($EI_{C3}$) | $EI_{C3} = E_{BF} \times \dfrac{2}{3} + E_{DS} \times \dfrac{1}{3}$ (2) | $E_{DS}$: Erodibility of dune system parameters |
| Erodibility of Dune System (3) ($E_{DS}$) | $E_{DS} = \dfrac{\frac{Pn_4 + Pn_5 + Pn_6 + Pn_7}{n_{Pn}} - 1}{A - 1}$ (3) | $Pn_4$: dune height <br> $Pn_5$ dune width <br> $Pn_6$: vegetation cover <br> $Pn_7$: washovers |
| Natural correction factor (4) ($CF_N$) | $CF_N = \dfrac{\frac{\frac{c_{1a}}{2} c_{1b} + c_2 + c_3 + c_4}{n_C} - 1}{A - 1}$ (4) | $c_{1a}$: significant wave height <br> $c_{1b}$ angle of wave approach <br> $c_2$: tidal range <br> $c_3$: sea level rise <br> $c_4$: storm surge |
| Sensitivity Index to natural processes (5) ($SI_N$) | $SI_N = EI \times \dfrac{3}{4} + CF_N \times \dfrac{1}{4}$ (5) | |

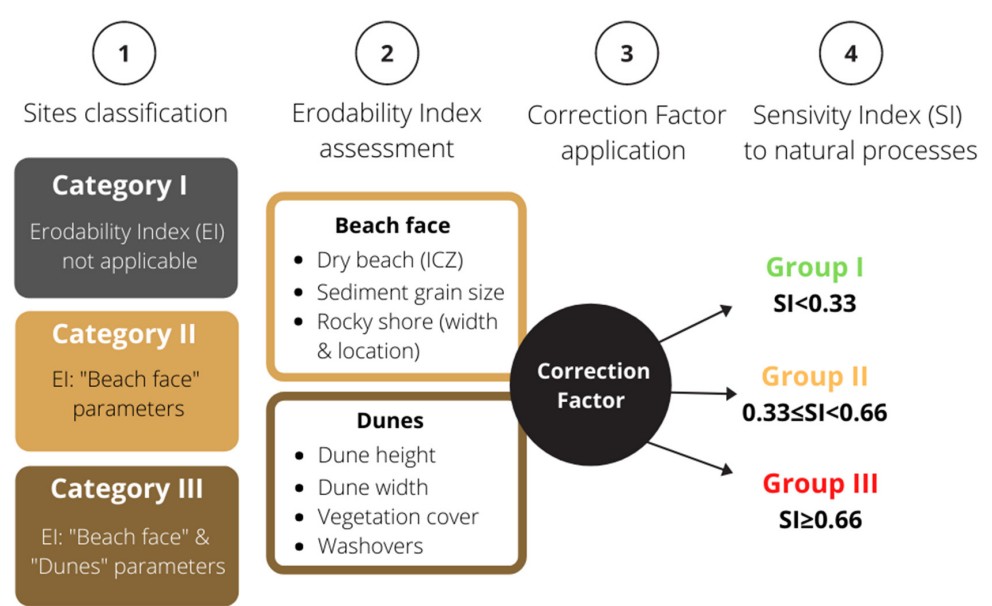

**Figure 5.** Steps for assessing scenic sensibility to natural processes.

Scores of EI (Categories II and III), CF and SI obtained, respectively, by Equations (1)–(5) are presented in a 0–1 range of values; "0" meaning very low scenic sensibility and "1" very high sensibility (Table 2).

### 3.2.1. Determination of Categories Concerning Natural Processes

Natural parameters considered by the CSES method were analysed in order to establish which are the most affected by marine processes, e.g., the erosion of a cliff sector does not mean its disappearance (points 1–3, Table 1). Four out of the eighteen physical parameters used in the CSES method were chosen, three belonging to the "Beach face" parameter (beach face type, width and colour, points 4–6, Table 1), and one corresponding to "Dunes" (point 10, Table 1), since erosion processes can strongly reduce beach width (point 5) or even favour beach and dune disappearance (points 4–6 and 10, Table 1). Further, according to Williams et al. [45] and Rangel-Buitrago and Anfuso [46], checklist matrices and sensitivity indices must be based on easily obtained information at any given area (or gathered mainly during the field observations), without requiring the analysis of exhaustive datasets. As a result, in a first step (Figure 5), according to the presence/absence of "Beach face" (points 4–6, Table 1) and "Dunes" (point 10) parameters, the sites investigated were classified among one of the pre-established categories:

- Category I. "No sensitive" sites (neither "Beach face" or "Dunes" parameters are present), e.g., Arrecife de las Sirenas at Cabo de Gata in Almeria province (Figure 6A) or rocky shore platform close to Ensenada del Tolmo at Gibraltar Strait in Cadiz province, Andalusia, Spain (Figure 6B). No further investigation regarding their sensitivity to natural processes is required.
- Category II. "Sensitive" sites that show "Beach face" but no "Dunes" parameters, e.g., Cantarrijan (Figure 6C) or pebble beaches, such as Cala del Pino (Figure 6D), both in the Maro-Cerro Gordo protected area in Granada province (Andalusia, Spain). To estimate each site's tradability value, the "Beach face" parameter is considered by taking into account beach width (as a multiple of the Imminent Collapse Zone, ICZ), sediment grain size and rock shore platform (width and location, Table 3).
- Category III. "Very sensitive" sites with "Beach face" and "Dunes" parameters (points 4–6 and 10; CSES), e.g., Los Genoveses or Barronal (Figures 5 and 6E,F), Almeria Province, Andalusia, Spain. The Erodibility Index is calculated by considering 2 sub-indexes: (i) "Beach face" (as above), and "Dunes" parameters, the latter considers dune height, width, vegetation cover and presence of washovers (Table 4). Sites with only remnants of dunes will be not considered as Category III (rated 2; point 10).

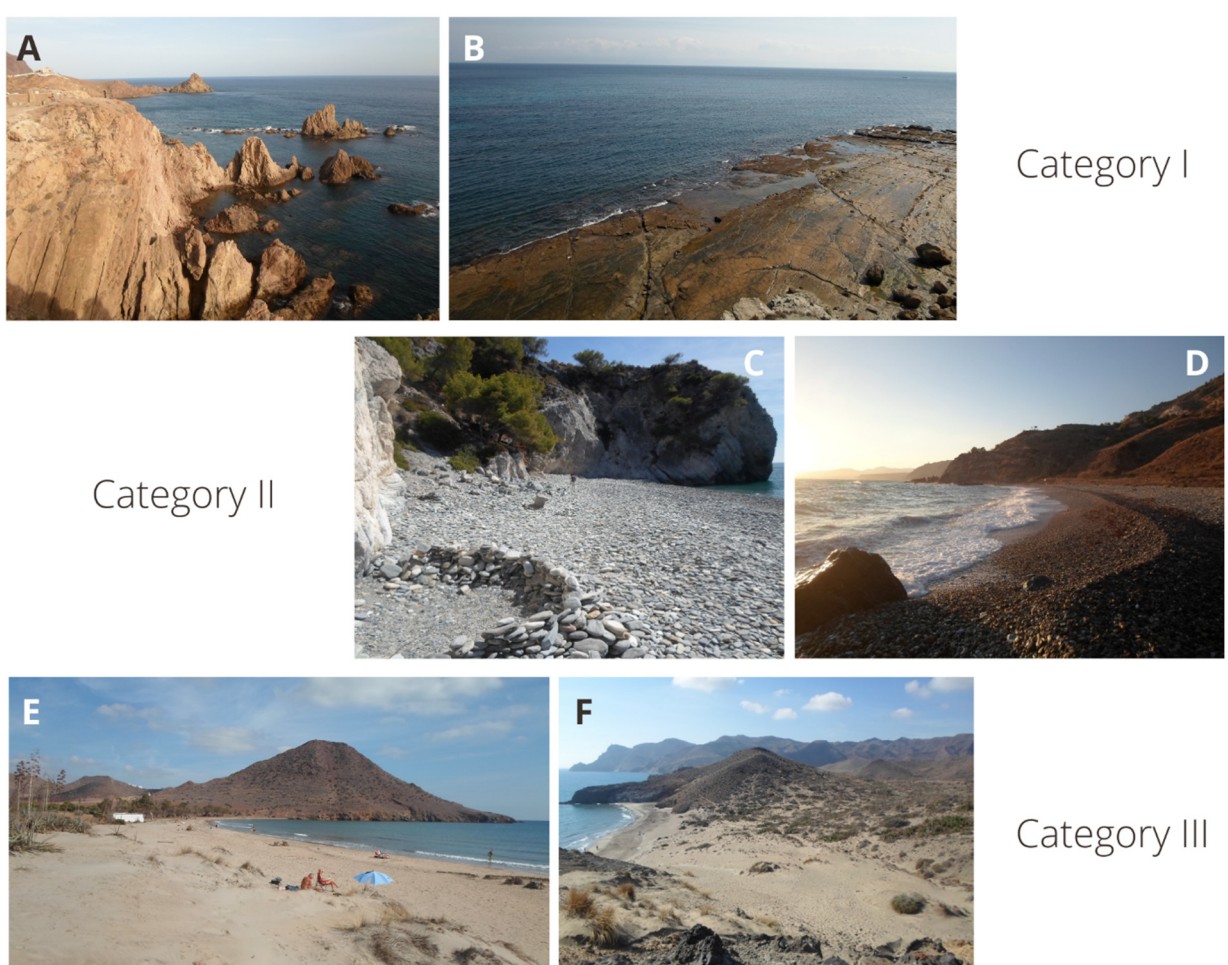

**Figure 6.** Examples of natural scenic categories in the Andalusia Mediterranean coast (Spain): Arrecife de Las Sirenas (**A**; Almería province) and rocky shore platform at Gibraltar Strait (**B**; Cadiz province); Cantarrijan (**C**; Granada province) and Cala del Pino (**D**; Malaga province); Los Genoveses (**E**) and Barronal (**F**) at Cabo de Gata in Almeria province.

#### 3.2.2. Determination of the Erodibility Index

In a second step, the Erodibility Index (EI) was used to calculate the level of sensitivity to natural processes of sites belonging to categories II and III by analysing the characteristics of "Beach face" and "Dunes" parameters (Tables 3 and 4, and Figure 5). The weight of each parameter considered was established according to the value used in the CSES method (Table 1) and presented in Equation (1) (Table 2) for Category II sites and in Equations (2) and (3) for Category III sites.

According to Rangel-Buitrago and Anfuso [46] and Rizzo et al. [47], all variables were classified on a 1–5 scale, 1 indicating a great contribution of a specific key variable to site resilience, 5 indicating a low contribution.

All physical parameters used have to be valued during field work observations and/or the consultation of existing maps/aerial photos/satellite images except the parameter "Dry beach width as a multiple of the Imminent Collapse Zone (ICZ)," which needs more detailed investigation since it is based on beach width and local retreat rates [48]. The description of the physical parameters used to determine "Beach face" and "Dunes" erodibility was presented in following manner:

(a) "Beach face" parameters

In order to semiqualitatively determine erodibility of sites classed in Category II, the

following 4 physical parameters (see details in Table 3 and Figure 5) were chosen since the presence/absence and characteristics of these parameters determine the erodibility of a beach environment.

(1)   Dry beach width as a multiple of the ICZ
      The ICZ is the area subject to imminent erosion, adjacent to the coastline, within a landward distance equal to 10 feet (3 m) plus five times the average annual erosion trend [48]. Predictions of the future coastline position can be based on coastal changes which have occurred over recent decades [48]. In detail, according to [49,50], shoreline position records calculated for medium- (10–60 years) or long-term (>60 years) time spans, usually by means of aerial photos and satellite images, are needed to derive a reliable trend because the longer is the investigated time span the lower is the effect of seasonal variations and the influence of individual storms on shoreline evolution [51,52].

(2)   Sediment Grain Size
      The size of gravel and boulders restricts their transport and hence a boulder or a cobble coast is much more stable than a sandy beach [53]. In this paper, assessment values regarding sediment size relevance were modified from [54]. A rate of 3 refers to the coexistence of sediments ranging from fine sand to pebbles and, at places, to the presence of a "seagrass berm", a common feature composed by Posidonia's debris on Mediterranean beaches.

(3,4)  Rocky shore width and location
      Is common to observe a rocky shore platform at the base of most cliffs or in front of many beaches, at about the mid-tide elevation location; this form is also called wave-cut platform or wave-cut bench. Such surfaces may measure from a few metres to hundreds of metres [55]. They are the results of long erosion processes and, at the same time, represent a natural defence for beaches, dissipating wave energy and reducing cliff's recession rates [55–57]. Two parameters were chosen to assess the resilience of rock-shore platforms:

  (i)   Width
        Understanding the relationship between platform width and exposure of wave energy is complex [53]. The lack of a clear relationship reflects the strong influence that geology has in determining morphological characteristics. However, it is clearly demonstrated that increasing width would attenuate wave energy and, hence, reduces back beach/dune erosion rate. Trenhaile [58] considers that the width of rocky shore is determined by the intensity of the erosional processes, the resistance of the rocks, and the length of time that the processes have operated. Width values presented herein were modified from [58].

  (ii)  Location
        According to its location, e.g., in the foreshore or dry beach, the rocky shore effects on wave energy is different; in this case the values proposed by [56] were considered.

(b)   "Dunes" parameters
      Dunes constitute natural sea defence, and a synergy exits between them and the beach, the dunes acting as a sediment reservoir to sustain the natural dynamic equilibrium between erosion and accretion processes. They also constitute a relevant aesthetic component in scenery assessment. Thus, in order to determine the erodibility of sites with beach and dunes parameters (Category III), 8 physical parameters were chosen and divided into 2 subindex categories: 4 "Beach face" parameters (previously described for Category II) and 4 "Dunes" parameters (Table 4 and Figure 6).

(1)   Mean Dune Height
      This was evaluated by considering dune crest height—classified following

the range proposed by Gracia et al. [59], from embryo dunes lower than 1 m height (rated 5) to a system superior to 6 m (rated 1).

(2) Mean Dune Width
Classified following the values proposed by Gracia et al. [59], width ranging from <25 m (rated 1) to >100 m (rated 5). This parameter can be evaluated during field surveys or detailed available topographic maps/models.

(3) Vegetation Succession Continuity
This identifies the grade of development and ecological conservation of the dune systems, their resilience to erosion/flooding processes and scenic value [60–62]. It can be determined mainly by field observations or the use of aerial photographs and satellite images.

(4) Percentage of washovers
Dune ridge continuity is often interrupted by washover fans that constitute hot spots sensible to coastal erosion and are linked to washover processes that can greatly affect the capacity of resilience of this system; in fact, if the dunes are eroded or fragmented, their resilience and storm-protection function are greatly dismounted or even totally lost [63–65].

**Table 3.** "Beach face" parameter rating.

| Parameter | | Null/Very low (1) | Low (2) | Medium (3) | High (4) | Very High (5) |
|---|---|---|---|---|---|---|
| Dry Beach as a Multiple of the ICZ [48] | | Accretion/ >5 times ICZ | 4 times ICZ | 3 times ICZ | 2 times ICZ | ≤ICZ |
| Sediment Grain Size modified from [54] | | Gravel/pebbles | | Medium/ coarse sand or mixed | | Fine sand |
| Rocky Shore | Width (m) mod. from [58] | >80 | 80–60 | 60–40 | 40–20 | <20 |
| | Location [56] | Nearshore | | Foreshore | | Absent |

**Table 4.** "Dunes" parameters rating.

| Parameter | Null/Very low (1) | Low (2) | Medium (3) | High (4) | Very High (5) |
|---|---|---|---|---|---|
| Dune Height (m) [59] | ≥6 | ≥3 | ≥2 | ≥1 | <1 |
| Dune Width (m) [59] | >100 | >75 | >50 | >25 | <25 |
| Vegetation cover [59] | Complete with fixed dune (forest) | Complete with fixed dune (shrub) | Semicomplete (without fixed dune) | Semicomplete (without embryo dune) | Incomplete or absent |
| Washovers (%) [65] | 0 | ≤5 | ≤25 | ≤50 | ≥50 |

### 3.2.3. Determination of the "Correction Factor" for Natural Processes

In a third step, a Correction Factor (CF) was considered to take into account the actual forcing variables, which can favour site erosion and, likely, the effects of future Climate Change trends (Equations (4) and (5) in Table 2).

Forcing variables can be defined as the level of potential stress that a given area could experience from an extreme event or the constant action of energetic events. Coastal experts consider waves as the most dominant force causing coastal erosion [66]. Considering the actual data availability at European scale, in order to calculate the Correction Factor, this paper considered the following 3 variables and trends (Table 5).

(a) Forcing variables

(1) Waves' characteristics

      (i)      Significant Wave Height

This is traditionally defined as the average wave height ($H_s$, from trough to crest) of the highest third (33.33%) of the waves in a given sample period, and in this paper it will be assessed following the model and intervals used by Jones and Monismith [67]. At the global level, the European Centre for Medium-Range Weather Forecasts (ECMWF) [68] provides a good database service with real-time and archive forecasts, analyses, climate reanalyses, reforecasts and multimodel datasets. In Spain, for example, wave data along the coast can also be obtained from the official online website "Puertos del Estado" ("Ports of the State") [69] through virtual buoys and stations collecting real-time weather [64]. Autonomous systems for specific regions, such SAPO –Autonomous Wave Forecast System of Balearic Islands [70]—can be used too for assessing this parameter.

      (ii)     Angle of wave approach to coastline

The degree of littoral exposition to wave fronts affects its sensitivity to storms impacts and erosion processes. It has been considered according to specific studies on sand coast exposure carried out by [65]. As for the determination of $H_s$, weather forecast reports can be used to assess the angle of wave approach.

(2)    Tidal Range

A high tidal range is associated with stronger tidal currents, which favour erosion and sediment transport [71,72]. Accordingly, macrotidal coasts (>4 m) will be more sensitive to erosion than those with smaller tidal ranges. However, a diversity of studies, e.g., [73–75], support the opposite view, i.e., a microtidal coastline is essentially always near high tide and, therefore, always at the greatest risk of significant storm impact. It is not unlikely that sea level during a storm surge event at macrotidal coasts is significantly lower than the high-tide level, which would increase the possibility of reduced flood risk [76]. This last approach is used in this paper and the tidal ranges used are "Microtidal" (<2 m), "Mesotidal" (2–4 m) and "Macrotidal" (>4 m) environments. Regarding the case studies in Andalusia, they are all located in a microtidal environment.

(b)  Regional trends

Regional trends, linked to global Climate Change processes, such as Sea Level Rise (SLR) and Storm Surge (SS) at regional scales are also used as correction factors. Studies, such as those of Morim et al. [77] and Vousdoukas et al. [78], agreed that an increase in planet temperature will also alter ocean waves along more than 50% of the world's coastlines with significant implications for coastal flooding and erosion processes. At a European scale, extreme water levels due to relative SLR can be further enforced by an increase in the extreme SS level, which can exceed 30% of the relative SLR, especially for the high return periods and pathway under the RCP8.5 scenario—the highest greenhouse gas emissions scenario established by the IPCC— [78]. In addition, an exhaustive review of existing papers and reports can be used for assessing both SLR and SS trends at different scales—see below.

(1)    Sea Level Rise (SLR)

A large and recent dataset on SLR trend is available from the Copernicus Programme [79] or ECMWF [68]. These databases provide data in absolute values that must be corrected, taking into account the local/regional subsidence and continent uplift values. In addition, a lot of the literature that deals with the SLR trend at different scales can be found in [69,80–82].

(2)    Storm Surge (SS)

Dataset on SS level can be found in the European Commission website through the application "Large Scale Integrated Sea Level and Coastal Assessment Tool" (LISCOAST) [83]. Dataset used from LISCOAST corresponds to the RCP4.5

scenario projected to 2099 and an extreme SS level associated with a 50 year return storm. Simulations carried out by previous authors do not include astronomical tidal components or projections of relative SLR [78].

**Table 5.** Correction Factor: forcing variables and trends.

| | Correction Factor | | Null/Very low (1) | Low (2) | Medium (3) | High (4) | Very High (5) |
|---|---|---|---|---|---|---|---|
| Forcing | Wave characteristics | Significant wave height (m) [67] | <0.75 | | 0.75–1.5 | | >1.5 |
| | | Angle of approach [65] | 10–45° (Oblique) | | 0–10° (Subparallel) | | 0° (Parallel) |
| | Tidal Range | | Macrotidal | | Mesotidal | | Microtidal |
| Trends at regional scale | Sea Level Rise (cm) * [13] | | <0 | | 0–40 | | >40 |
| | Storm Surge (m) ** [13] | | <1.5 | | 1.5–3 | | >3 |

\* Estimation expected by the end of the century (2100). \*\* Highest water level recorded and associated with storm events according to [13] scale.

### 3.2.4. Determination of the "Sensitivity Index"

In a fourth step, once the CF is calculated, a Sensitivity Index (SI) to coastal natural processes ($SI_N$) is achieved according to the standard equation (5, Table 2) whose scores are presented in 0–1 range of values, "0" meaning Null/Very low and "1" Very high contribution. Thereafter, sites are classified into one of the following 3 groups (Figure 5):

- Group I: scenic parameters not sensitive to natural processes, $SI_N < 0.33$. Category I sites belong to this group.
- Group II: sites with intermediate values at Erodibility Index and Correction Factor, with a Sensitivity Index $\geq 0.33$ and $< 0.66$.
- Group III: highly sensitive scenic parameters that require special attention from coastal managers, $SI_N \geq 0.66$.

### 3.3. Determination of Scenic Sensitivity to Human Pressure

Coasts represent very fragile environments affected by the disordered emplacement of infrastructures and activities, such as industry, massive tourism, agriculture, fishing, etc. Today, beaches are responsible for more than a half of tourism income across the world, and studies argued that a square metre of beach could produce up to EUR 12,000 per year [84]. Therefore, beach experts agree that managing this very complex and worthy coastal system should be done with the right powerful tools, seeking balance between economic development and sustainability. On the other hand, unsustainable growth not only affects coastal ecosystems but, in the medium term, also severely impacts the different benefits of tourism, modifying and destroying the sustenance of the tourism activity: Scenery. Rangel-Buitrago [33] (p. 5) remarked that "the challenge can be summarized in a very simple question that demands a smart answer: how to develop a 3S tourism that will not minimize the quality of the natural resource and benefits to stakeholders?"

The aim of this section is to determine the CSES anthropogenic aspects (points 19–26, Table 1) regarding sensitivity to human pressure in a scenario of increasing human activities and developments at the coastal zone (Figure 7). To this purpose, a Human Impact Index (HI), Correction Factor (CF) and Sensitivity Index (SI) for human pressures were calculated according to four steps and are described in detail in Table 6.

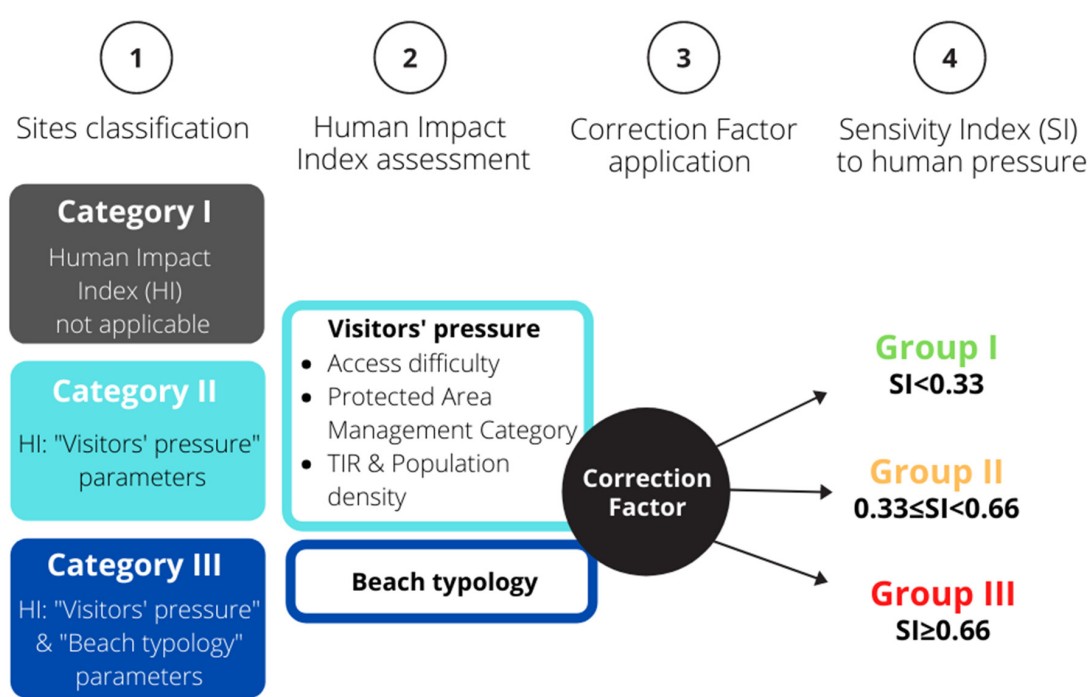

**Figure 7.** Steps for assessing scenic sensibility to human pressure.

**Table 6.** Equations regarding Human Impact Index (related to sites categories), Correction Factor and Sensitivity Index to human pressure.

| Indexes and Correction Factor | Equations | Parameters |
|---|---|---|
| Human Impact Index (6) for Category II sites ($HI_{C2}$) | $$HI_{C2} = \dfrac{\frac{Ph_1 + Ph_2 + \frac{Ph_{3a} + Ph_{3b}}{2}}{n_{Ph}} - 1}{A - 1} \quad (6)$$ | $Ph$: human parameter <br> $Ph_1$: access difficulty <br> $Ph_2$: protected area management category <br> $Ph_{3a}$: tourism intensity rate <br> $Ph_{3b}$: population density <br> $n_{Ph}$: number of human parameters (3) <br> $A$: maximum attribute value (5) |
| Human Impact Index (7) for Category III sites ($HI_{C3}$) | $$HI_{C3} = \dfrac{\frac{Ph_1 + Ph_2 + \frac{Ph_{3a} + Ph_{3b}}{2} + Ph_4}{n_{Ph}} - 1}{A - 1} \quad (7)$$ | $Ph_4$: beach typology |
| Human Correction Factor (8) ($CF_H$) | $$CF_H = \dfrac{c - 1}{A - 1} \quad (8)$$ | $c$: tourism trend |
| Sensitivity Index (9) to human processes ($SI_H$) | $$SI_H = HI \times \frac{3}{4} + CF_H \times \frac{1}{4} \quad (9)$$ | |

### 3.3.1. Determination of Categories Concerning Human Impacts

In a first step (Figure 7), the Human parameters in Table 1 were analysed to determine their relationship with human processes/actuations and the protection feature of a site. A large number of visitors could directly affect "Noise disturbance", "Litter" and "Sewage discharge evidence" (points 19–21, Table 1). The quality of other parameters, such as

"Non-built environment", "Built environment", "Access type" and "Utilities" (points 22–24 and 26, Table 1), are mainly linked to land use and beach typology and therefore to the protection feature of the site (if any). "Skyline" (point 25, Table 1) refers to the level of urbanization of surrounding areas so it is not strictly related to the site's own characteristics or protection features. Therefore, sites investigated were classified among one of the pre-established categories (Figure 7):

- Category I. Sites with null human disturbance (all Human parameters of Table 1 have very good scores), e.g., Cala Chumbo at Malaga province, located in a Natural Place, which shows top scores (i.e., 5, Table 1) of all the human parameters (only accessible by boat or a 50 minutes' walk from the nearest car parking, Figure 8A); another example is Punta Sabinar, located in a strict natural reserve at Almeria province. No further investigation regarding their sensitivity to human pressure is required since they are usually located in very natural and isolated areas and/or are under a strong protection feature [29,41].
- Category II. Sites that show very low human impacts mainly linked to "Noise disturbance", "Litter", "Sewage discharge evidence", and of temporally emplaced elements such as litter bins, sun loungers, beach umbrella within the "Utilities" parameter (points 19–21, 26 Table 1), which are essentially related to human affluence and tourist season; e.g., Cala Arena or Ensenada del Tolmo at Cadiz province show high values at human parameters except at "Litter" (Figure 8B). Such sites are usually located in natural areas under any protection feature [29,41].
- Category III. Sites that show some human impacts reflected by medium scores at "Noise disturbance", "Litter", "Sewage discharge evidence" but also at "Non-built environment", "Built environment", "Access type", "Skyline" and "Utilities" (points 22–26, Table 1), and linked to land use and beach typology and, therefore, to the protection feature of the site (if any); e.g., Cala El Cañuelo located in a rural environment at Malaga province (Figure 8C). Such sites are usually located in (or at the border of) natural areas, with a low level of protection feature or none [29,41].

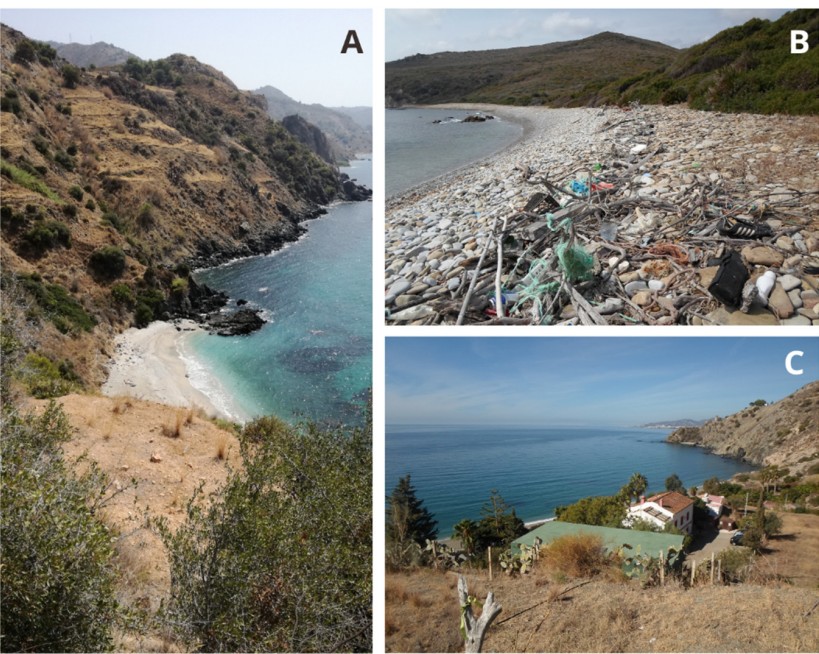

**Figure 8.** Examples of human scenic categories in Andalusia Mediterranean coast (Spain): Cala Chumbo at Malaga province (Category I, **A**); Ensenada del Tolmo at Gibraltar Strait in Cadiz province (Category II, **B**); Cala El Cañuelo at Malaga province (Category III, **C**).

### 3.3.2. Determination of the "Human Impact Index"

In a second step (Figure 7), concerning the Human Impact Index (HI, Table 6), the following two variables, which have a side effect on the eight human parameters of the CSES method, were considered: (1) "Visitors' pressure", which included "Access difficulty", "Protected Area Management Category", "Tourism Intensity" and "Population Density" parameters (for Category II sites); (2) "Beach typology" (for Category III sites), (Figure 7, Tables 7 and 8).

**Table 7.** Parameters used to assess Human Impact Index.

| Parameter | | Null/Very Low (1) | Low (2) | Medium (3) | High (4) | Very High (5) |
|---|---|---|---|---|---|---|
| Access difficulty (min.) [29] | | >45 or only by sea | 25–45 | 10–25 | 5–10 | <5 |
| Protected Area Management Category [86] | | Ia—Ib | II—III | IV-V-VI | Only local designation | No |
| Tourism Intensity Rate (TIR) and Population density (PD) | TIR: tourist beds 1000 inhab. mod. from [87] | <200 | | 200–500 | | >500 |
| | PD: persons/km$^2$ mod. from [90] | <100 | | 100–700 | | >700 |

**Table 8.** "Beach typology" rating.

| Parameter | Null/Very Low (1) | Low (2) | Medium (3) | High (4) | Very High (5) |
|---|---|---|---|---|---|
| Beach typology [23] | Remote | | Rural | | Village or Resort |

As stated for natural systems, variables were classified on a 1–5 scale, and scores of Human Impact Index, Correction Factor and Sensitivity Index are obtained by the Equations (6)–(9) and presented in 0–1 range of values with the same standard previously defined. A description of the parameters used to determine human pressure is presented in following lines.

(a) Visitors' pressure

Coastal areas are generally the most popular during the summer months. As mentioned previously, pressure due to a high concentration of visitors, in a short period of time, comes with a lot of impacts on fragile coastal ecosystems. Regarding landscape, a large number of visitors directly affect 4 of the 8 components of the CSES parameters, i.e., "Noise disturbance", "Litter", "Sewage discharge evidence" and "Utilities" (points 19–21, 26, Table 1), and this essentially depends on the following two parameters:

(1) Access difficulty

The facility of access strongly contributes to a site's affluence, and sites considered as "remote" demand at least a walk up to 300 m or more therefore distance to walk is a first indicator of access difficulty [23]. The approach proposed in this paper considered the access difficulty according to a difficulty scale presented in Table 7 and used by [29]; e.g., a 300 m walk in a coastal cliff or mountainous context will be quite longer/harder than in a flat coastal plain. Further, a wide diversity of studies, e.g., [22,84], support that "Access difficulty" is an important part of management strategies to regulate and protect sites from overtourism. For example, in Andalusia (Spain), Mooser et al. [29] observed that 30% of sites ranked as Class I required at least a 25 minutes' walk,

and some were only accessible by sea. This parameter is assessed according to field work experience and time walking from the nearest car parking.

(2) Protected Area Management Category

Generally, protected coastal areas are sites of large interest for the conservation of natural habitats and ecosystems and of great relevance for research and educational purposes and sensitive tourism. However, it is inordinately complicated to understand the division and categorization of protected areas. In Europe, 685 designation types have been recorded across 39 countries (EEA, 2012). There are many types of site designation each one, showing specific objectives, spatial boundaries and specific governance policies. Certain spatial areas of high environmental value can be covered partly or totally by a number of different designations applied at local, regional, national or international level. An exhaustive analysis of this complexity can be found in the report "Protected Areas in Europe" [85]. It is considered that, among the 8 anthropogenic parameters of the CSES, 4 can be directly related to the management categories of protection: built and nonbuilt environment, access visibility and utilities. Given this context, the method proposed in this paper uses the standard methodology established by the International Union for Conservation of Nature [86], known as the Protected Area Management Category, which ranges from very strict to relatively permissive areas:

Ia: Strict Nature Reserve;
Ib: Wilderness Area;
II: National Park;
III: Natural Monument or Feature;
IV: Habitat/Species Management Area;
V: Protected Landscape/Seascape;
VI: Protected area with sustainable use of natural resources.

IUCN categories are considered among the best tools for providing a comparative picture of protected areas at an international scale. Finally, local designation type (e.g., at a council level) is also considered (rated 4).

(3) Tourist Intensity Rate (TIR) and Population Density (PD)

The global tourism industry is extremely concentrated in coastal areas where an increase in building and infrastructure has increased environmental pressure on protected, natural/seminatural territories, and local communities [15]. The following two parameters will be considered to characterize the level of potential stress to which each site may be exposed:

(i) Tourism Intensity Rate (TIR)

Known also as the tourism intensity indicator, the TIR is the ratio between the capacity of tourist accommodation in municipalities (number of tourist beds) and their permanent resident population. Quantifying the theoretical increase in the population in times of tourist influx is a clear indicator of tourism pressure [87–89].

The scale presented in Table 7 was modified from the Geostatistics Information System of Andalusia (SIGEA) [90] and [87]—which carried out this analysis at a local level in 6000 municipalities in mainland France and overseas territories—and has been divided into 3 different levels:

- TIR less than 200 beds per 1000 inhabitants.
- TIR between 200 and 500 beds per 1000 inhabitants.
- TIR > 500 beds per 1000 inhabitants.

For example, a TIR of 1000 means that the destination has a tourist capacity equivalent to the permanent population and the area is, therefore, likely to double its population during summer period.

      (ii)     Population density (PD)

"The most significant TIR are located in territories with a low annual population and a high accommodation capacity. However, there can also be significant tourism pressures on the environment without the tourism intensity rate being high." [87] (p. 8). This is notably the case for destinations with a high population density. Therefore, to complement the TIR, the density of population will be also considered following the standard presented in Table 7.

(b)    "Beach typology" parameter

Different beaches have different users and need different management strategies [91]. Beaches cannot and should not be compared as a whole, but rather considered by their respective typologies. Indeed, on remote and natural beaches, users considered the scenic aspect to be of most importance while, in urban beaches, facilities and car parking aspects were deemed to be the prime factors [91]. Beach rewards, such as the famous Blue Flag, are almost only based on anthropogenic concepts, services proximity, leisure, etc., and not on natural aspects, such scenic beauty, and peace and quiet—showing the limit of the present-day management approach in a coastal context [92,93]. Williams [23] categorized five beach types: resort, urban, village, rural and remote, and proposed the following definitions:

- Remote: These may be defined by difficulty of access, largely by boat or on foot—a walk of up to 300m+. They can be adjacent to either villages or rural areas but rarely with urban areas. They are not supported by public transport and have very limited (<5 if any) temporary summer houses.
- Rural: These would be found outside the urban/village environment. They are not readily accessible by public transport and have virtually no facilities—perhaps a small summer shop, car park and/or toilet. In the Mediterranean context, permanent land-based recreational amenities (such as golf courses) and summer time beach-related recreational facilities (e.g., banana boats, jet skiing etc. which are typical of resorts) may be found.
- Village: These are found outside the main urban environment. They have a small, permanent population reflecting access to organised but small-scale community services (such as a primary school(s), religious centre(s) and shop(s). This also includes "tourist villages" mainly utilized in the summer months as well as "ribbon development" between urban and rural environments.
- Urban: These serve large populations, which have well-established public services, e.g., primary school(s), bank(s), religious centre(s), internet cafes, with a clearly demarcated central business district, and commercial activities, e.g., harbours and marinas. Urban beaches are located within/adjacent the urban area and are, in the main, freely open to the public.
- Resort: Especially in tourist "hot spots" e.g., the Caribbean, Mediterranean, coastal resorts should be located on a beach adjacent to an accommodation complex (hotel/apartment/camp site), where a substantial proportion of beach users are residents and management is the responsibility of the complex. A host of facilities is usually prevalent, e.g., wind surfing, speed boat towing activities (e.g., "rings", "bouncy castles", "bananas") as recreation is the main aim; Club Med epitomises this type of resort.

Therefore, any sound and effective management plan should always take cognisance of typology, and this is a remarkable aspect to determine tourism pressure assessment, and this was the case in this paper. From a scenic perspective, it is quite obvious that pressure is not the same in a resort area as in a remote place. Coastal scenic sites ranked in Class I or Class II by means of the CSES method, are usually located in remote and rural areas and, in some special cases, in a village and/or resort locations. "Beach typology" can affect almost every CSES parameters and, particularly, "Built" and "Non-built" environment, "Access

type", "Utilities" (points 22, 23, 24 and 26) that characterize Category III sites. Ratings are presented in Table 8.

### 3.3.3. Determination of the "Correction Factor" for Human Pressure

International Tourist Arrivals (ITA) is expected to increase worldwide by 65% from 2010 to reach a number of 1.8 billion arrivals per year by 2030 [94]. While Europe will remain the dominant region with almost 780 million tourists, ITA will increase by 150% for Asia and the Pacific regions, by 50% in America, and will double in Africa by 2030 [94]. Coastal tourism has become a major economic sector for countries with accessible and attractive coastlines but, at the same time, strongly contributed to its fragilization putting at risk its own sustainability. As stated Tonazzini et al. [95] (p.11) "emerging destinations are becoming emergency territories due to its rising vulnerability to environmental risks, in particular, climate change, dependency to natural resources, quality of ecosystems, putting in danger coastal territories and local communities and requesting urgent policy answers". According to the Eurobarometer on the European tourism preferences [15], in France, as in the rest of Europe, the attractiveness of a natural area is the first criterion for tourists to return to holiday in the same place. On a global basis, pristine coastal tourist destinations, with great scenic beauty, can be contemplated as emerging destinations with increasing demand. The changes in population caused by tourism raise the question of a destination's carrying capacity, and the need to identify the level or threshold of tourist traffic that must not be exceeded to not compromise the "health" of a destination's environment.

In this context, it is essential to consider the medium and long-term tourism trends/patterns to anticipate growth scenarios, and increased resilience of coastal landscape beauty. Indeed, the phenomenon of soil degradation resulting from increased urbanisation, and/or massive influxes of visitors, can considerably affect several parameters of the CSES method.

After considering tourism intensity and population density indicators for assessing HI, the method will consider, as a single correction factor, the tourism trend at investigated sites. For this purpose, data need to be collected at a very local level (coastal municipalities or NUTS 5) since national and regional averages bear the risk of presenting a misleading picture in provinces, where there exist significant disparities between areas and municipalities. For this study, tourism trends have been analysed according to the evolution of the number of beds in tourist establishments (during the last 15 years) according to the rating presented in Table 9.

**Table 9.** Human Correction Factor: tourism trend at local scale.

| Correction Factor | Null/Very Low (1) | Low (2) | Medium (3) | High (4) | Very High (5) |
|---|---|---|---|---|---|
| Evolution of the number of beds in tourist establishments (%) * mod. from [90] | <5 | 5–20 | 20–40 | 40–60 | >60 |

\* during at least the last 10-year period.

### 3.3.4. Determination of the "Sensitivity Index" to Human Pressure

Once we calculated the CF, using Equation (8), an *SI* to coastal human processes ($SI_H$) will be achieved, according to Equation (9). Sites could be classified into one of the following three groups, according to the same standard previously established to determine the sensitivity to natural elements (Figure 7):

Group I, $SI_H < 0.33$;
Group II, $0.33 \geq SI_H < 0.66$;
Group III, $SI_H \geq 0.66$.

*3.4. Determination of the total Sensitivity Index to Natural Processes and Human Pressure*

Once the sensitivity of a site to natural processes and human pressure has been determined, a total Sensitivity Index (SI) is calculated and given in Equation (10):

$$SI = \frac{SI_N + SI_H}{2} \tag{10}$$

As stated for natural and human sensitivity indexes, scores obtained by the mentioned equation are presented in 0–1 range of values ("0" meaning Null/Very low and "1" Very High Contribution), allowing us to classify sites among one of three established sensitive groups:

Group I, $SI < 0.33$;
Group II, $0.33 \geq SI < 0.66$;
Group III, $SI \geq 0.66$.

Finally, for wide-scale studies, it could be interesting to represent, in a map, the different sensitivity groups, making the results and priorities easier to read and interpret for coastal managers, e.g., [96]. An overview of parameters, trends, correction factors and sources used for sensitivity assessment regarding natural processes and human pressure can be found in Appendix A.

## 4. Results and Discussions: Case Studies from Andalucía

The method proposed in this study was applied to 29 coastal sites of great scenic beauty, which were described in detail by Mooser et al. [29], belonging to classes I (22) and II (7) of the CSES method, and located along the Mediterranean coast of Andalusia. The main characteristics and results of the scenic sensibility analysis are presented in Table 10 and Figure 9, which gives a visual view on the relation between *SI* and "D" value (CSES) to determine priorities in terms of policies and management.

**Table 10.** Main characteristics of sites studies in the Andalusia Mediterranean coast (Spain) classified by municipalities and provinces: CSES indexes (D), natural and human categories (NC, HC), Erodibility and Human Impact Index (EI, HI), sensitive groups (G) and Sensitivity Indexes (SI).

| Sites | Municipality | Prov. | CSES "D"/Class | Natural Processes | | | | Human Pressure | | | | Total SI | Group |
|---|---|---|---|---|---|---|---|---|---|---|---|---|---|
| | | | | NC | EI | SI | G | HC | HI | SI | G | | |
| 1. Ensenada del Tolmo | Tarifa | CA[1] | 0.84; II | II | 0.17 | 0.27 | I | II | 0.33 | 0.37 | II | 0.32 | I |
| 2. Cala Arena | | | 0.91; II | II | 0.50 | 0.52 | II | II | 0.42 | 0.44 | II | 0.48 | II |
| 3. Cabopino | Marbella | | 0.73; II | III | 0.44 | 0.53 | II | III | 0.75 | 0.63 | II | 0.58 | II |
| 4. Punta de Vélez * | Velez | | 0.77; II | II | 0.50 | 0.55 | II | III | 0.62 | 0.53 | II | 0.54 | II |
| 5. Caleta de Maro | | | 0.84; II | II | 1.00 | 0.92 | III | III | 0.56 | 0.55 | II | 0.73 | III |
| 6. Cala Chumbo | | MA[2] | 1.02; I | II | 0.46 | 0.52 | II | I | 0.0 | 0.0 | I | 0.26 | I |
| 7. Las Alberquillas | Nerja | | 0.95; I | II | 0.50 | 0.55 | II | II | 0.5 | 0.50 | II | 0.52 | II |
| 8. Cala del Pino | | | 0.95; I | II | 0.33 | 0.42 | II | II | 0.5 | 0.50 | II | 0.46 | II |
| 9. Cala El Cañuelo | | | 0.90; I | II | 0.50 | 0.55 | II | III | 0.5 | 0.50 | II | 0.52 | II |
| 10. Cantarrijan | Almuñecar | | 1.07; I | II | 0.33 | 0.42 | II | III | 0.56 | 0.61 | II | 0.51 | II |
| 11. Cala El Cambron | Salobreña | GR[3] | 0.68; II | II | 0.33 | 0.42 | II | III | 0.68 | 0.64 | II | 0.53 | II |
| 12. La Rijana | Gualchos | | 0.89; I | II | 0.38 | 0.45 | II | III | 0.62 | 0.47 | II | 0.46 | II |
| 13. El Ruso * | Albuñol | | 0.96; I | II | 0.67 | 0.67 | III | II | 0.5 | 0.38 | II | 0.52 | II |

**Table 10.** *Cont.*

| Sites | Municipality | Prov. | CSES "D"/Class | Natural Processes | | | | Human Pressure | | | | Total SI | Group |
|---|---|---|---|---|---|---|---|---|---|---|---|---|---|
| | | | | NC | EI | SI | G | HC | HI | SI | G | | |
| 14. Punta Sabinar | El Ejido | | 0.82; II | II | 0.83 | 0.83 | III | I | 0.0 | 0.0 | I | 0.42 | II |
| 15. Cala Arena | | | 0.96; I | II | 0.79 | 0.73 | III | II | 0.33 | 0.37 | II | 0.55 | II |
| 16. Cala Raja | | | 1.04; I | II | 0.79 | 0.73 | III | II | 0.33 | 0.37 | II | 0.55 | II |
| 17. Cala de la Media Luna | | | 1.01: I | II | 1.00 | 0.89 | III | II | 0.42 | 0.44 | II | 0.67 | III |
| 18. Monsul | | | 1.19; I | III | 0.44 | 0.47 | II | II | 0.42 | 0.44 | II | 0.46 | II |
| 19. Barronal | | | 1.03; I | III | 0.67 | 0.64 | II | II | 0.33 | 0.37 | II | 0.51 | II |
| 20. Cala Grande | Nijar | | 1.09; I | II | 1.00 | 0.89 | III | II | 0.25 | 0.31 | I | 0.60 | II |
| 21. Los Genoveses | | | 1.26; I | III | 0.76 | 0.71 | III | II | 0.42 | 0.44 | II | 0.58 | II |
| 22. El Playazo | | AL[4] | 1.12; I | II | 0.67 | 0.64 | II | II | 0.42 | 0.44 | II | 0.54 | II |
| 23. Cala de San Pedro | | | 0.85; I | II | 1.00 | 0.89 | III | III | 0.25 | 0.31 | I | 0.60 | II |
| 24. Cala del Plomo | | | 0.91; I | II | 0.92 | 0.89 | III | III | 0.44 | 0.46 | II | 0.67 | III |
| 25. Cala de Enmedio | | | 1.20; I | II | 0.58 | 0.64 | II | II | 0.25 | 0.31 | I | 0.48 | II |
| 26. Los Muertos | Carboneras | | 0.93; I | II | 0.50 | 0.58 | II | III | 0.25 | 0.44 | II | 0.51 | II |
| 27. El Sombrerico | Mojacar | | 0.88; I | II | 0.42 | 0.52 | II | III | 0.56 | 0.48 | II | 0.50 | II |
| 28. Bordenares | | | 0.94; I | II | 0.58 | 0.64 | II | III | 0.56 | 0.48 | II | 0.56 | II |
| 29. Los Cocedores * | Pulpi | | 0.83; II | III | 0.78 | 0.79 | III | III | 0.62 | 0.72 | III | 0.75 | III |

\* Sites not located in protected areas; [1] Cádiz; [2]: Malaga; [3]: Granada; [4]: Almeria.

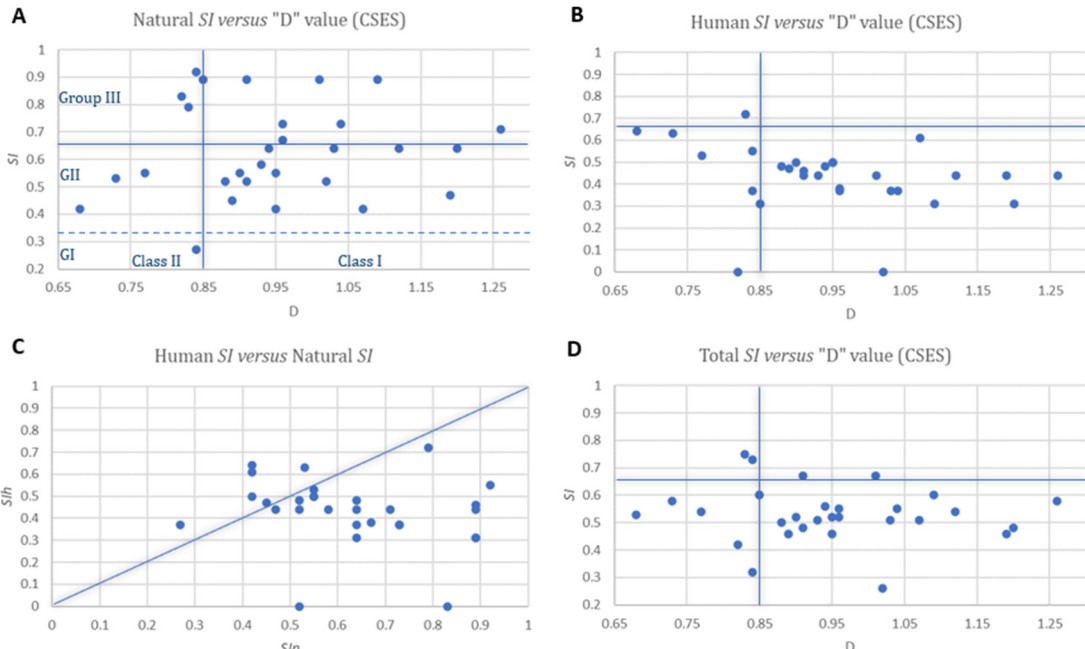

**Figure 9.** Natural *SI* versus "D" value (CSES) (**A**); Human SI versus "D" value (**B**); Total *SI* versus "D" (**C**) and Human *SI* versus Natural *SI* (**D**). Limits have been established according to Class I (D $\geq$ 0.85, CSES) and Group III (*SI* $\geq$ 0.66).

### 4.1. Sensitivity to Natural Processes

Regarding the natural categories, 24 sites were included in Category II and 5 in Category III. Only one place (i.e., Ensenada del Tolmo, Cadiz province) showed scenic parameters not sensitive to natural processes (*SI*: 0.27; Group I), whereas 17 were classified in Group II and 11 sites in Group III as highly sensitive scenic sites that require special attention from coastal managers (*SI* ≥ 0.66) (Table 10, Figure 9A). Among them, sites such as Caleta Maro (*SI*: 0.92) or Los Genoveses (*SI*: 0.71) stand out from the rest; the first one for its index value, the highest one, and the latter since it is one of the most attractive beaches in Andalusia and in Europe, according to other studies carried out with the same method.

Concerning the Erodibility Index calculation, "Beach face" and "Dunes" parameters were assessed during the field visits. For the correction factor, wave characteristics were obtained from Molina et al. [31], which used data modelled by ECMWF. The highest significant wave-height values were registered in the Granada province (0.92 m; rated 3), with an angle of approach parallel to the coast (rated 5). Regional trends of SLR and SS were, respectively, established according to the Spanish Government Report [97] and the LISCOAST dataset [78,83]. Intervals of SS projection were modified according to the small scale of the investigated area, located in a microtidal environment, and divided into the following classes (m): (1) < 0.80; (3) 0.80–1.00; (5) > 1.00. Sites most exposed to extreme SS level are located in the eastern coast of Almeria province (i.e., from Cala del Plomo to Cala Cocedores) with values superior to one metre (1.03; rated 5). All investigated sites showed >40 cm (e.g., score 5) ratings at relative SLR projections.

### 4.2. Sensitivity to Human Pressure

In total, 14 sites were included in Human Category II, 13 in Category III and only 2 sites in Category I; i.e., Cala Chumbo at Malaga province and Punta Sabinar at Almeria. Both places were not further investigated. All of the sites (but three) were under any feature of protection (Table 10) and just one site (i.e., Cabopino, Malaga province) showed the Blue Flag Award. Besides the two Category I sites (*SI*: 0.0), other two locations were assessed in Group I (Table 10, Figure 9B): Cala de Enmedio and Cala Grande in Almeria province at Cabo de Gata Natural Park (*SI* < 0.33). Cala Cocedores, a location without any kind of protection feature and with easy access, is the only site included in Group III (*SI*: 0.75, Table 10 and Figure 9B).

Regarding the Human Impact Index assessment, "Access difficulty" and "Beach typology" have been assessed according to field work observations. More than half of the sites required at least a 10 min walking from nearest car parking (attributes ≥ 3), and three were only accessible by boat or by a walk longer than 45 min (e.g., Ensenada del Tolmo at Cádiz or Cala Chumbo in Málaga province). All of the locations were located in remote and rural areas, except two sites: Cabopino (village) and Cala El Cambron (resort), respectively, in the Malaga and Granada provinces. "Protected Area Management Category" was established after reviewing the Plan of Management of Natural Recourses (*Plan de Ordenación de los Recursos Naturales, PORN,* in Spanish) and the Master Plan of Administration and Use (*Plan Rector de Uso y Gestión, PRUG,* in Spanish), both being defined by the Act 4/89. Concerning "Tourist Intensity Rate" and "Population Density", both were assessed by means of SIGEA [90]. The highest values of TIR were registered in the municipalities of Tarifa in the Cadiz province (508 beds per 1000 inhabitants) and Mojacar in Almeria (1700 beds), while the highest scores of "Population density" were clearly located in Malaga, particularly, in Marbella (1208 persons per square km$^2$). Finally, the trend of tourist establishments and number of beds was studied during the period 2004–2017 by means of the data presented by SIGEA [90]. Most of the investigated municipalities showed low to medium scores of the correction factor, with increasing numbers varying from 8% to 39%. Sites in Granada, Albuñol and Gualchos, showed negatives scores, whereas two municipalities stood out from the rest; i.e., Carbonera and Pulpi in the Almeria province with, respectively, 66.2% and 74% rising values.

Scores obtained for Natural *SI* and Human *SI* were compared in Figure 9 C highlighting as the investigated sites are quite more sensitive to natural processes than human pressure.

*4.3. Total Sensitivity Index, SI versus "D" value (CSES) and Natural SI versus Human SI*

Combining the results of sensitivity indexes obtained, respectively, for natural processes and human pressure, four sites drew attention from the rest: Caleta de Maro (*SI*: 0.73), Cala de la Media Luna (*SI*: 0.67), Cala del Plomo (*SI*: 0.67) and Cala Cocedores (*SI*: 0.75), all of them included in Group III (Table 10 and Figure 9D). The latter site requires specific and careful attention from coastal managers, since it showed very high sensitivity to natural and human aspects (Figure 9A,B).

**5. Conclusions**

In this paper, a Sensitivity Index of natural parameters to erosion processes was determined according to an Erodibility Index, calculated according to sensitivity of different natural parameters of the CSES method to coastal erosion, and a correction factor, obtained, taking into account waves, tidal range and sea level and storms' trends at local scale in a climate change context. A Sensitivity Index to human pressure/activities was determined according to a Human Impact Index, based on effects of visitors and site protection status on one human parameter of the CSES method, and a correction factor was obtained, taking into account local trends of tourism pressure.

It is possible to say that worldwide pristine landscapes in coastal areas are turning into emerging tourist destinations. Managers need sound tools to consider the medium- and long-term tourism trends/patterns to anticipate growth scenarios in order to increase the resilience of coastal scenic sites to massive influxes of visitors and urbanisation processes. The results obtained in Andalusia, South of Spain, allowed for the determination of the most sensitive sites to natural processes and human pressure, this way permitting the establishment of sound management tools. For examples, sites such Caleta de Maro (Malaga province) or Los Genoveses (Almeria province) require specific attention to erosion processes, while locations such as Cabopino (Malaga) need specific attention on human pressure. In some situations, both issues may demand the same attention, e.g., Los Cocedores, in Almeria province.

Therefore, considering the increasing importance of coastal tourism and especially of environmental and sustainable tourism, the method proposed in this paper is of large interest to management in order to prevent and limit environmental degradation, linked to natural processes and tourism pressure/developments and, if possible, to enhance scenic beauty by means of sound management strategies.

**Author Contributions:** A.M. and G.A. designed the study and participated in all phases. A.T.W. and P.P.C.A. provided a global structural discussion and participated in the development of the methodological approach. R.M., A.M. and G.A. carried out the field work observations and analysed the results. P.P.C.A. made contributions regarding the conceptual approach. All authors have read and agreed to the published version of the manuscript.

**Funding:** First author, A.M., is supported by a PhD scholarship under the program "Environmental Phenomena and Risk", cycle 35 of the *Università degli Studi di Napoli Parthenope* (Naples, Italy).

**Acknowledgments:** This work is a contribution to the Andalusia Research Group RNM-328 (University of Cádiz, Spain).

**Conflicts of Interest:** The authors declare no conflict of interest.

## Appendix A

**Table 1.** Summary of indexes, parameters, trends, correction factors and sources used for sensitivity assessment.

| Sensitivity Indexes | Indexes and Correction Factor (CF) | Parameters and Trends | | Assessment Method | Reference |
|---|---|---|---|---|---|
| Natural processes | Erodibility Index | Beach face | Dry Beach as a Multiple of the ICZ | Aerial photographs and satellites images | [48] |
| | | | Sediment Grain Size | Field work observations * | [54] |
| | | | Rocky shore width and location | | [56,58] |
| | | Dunes (if Category III sites) | Dune Height | Field work observations * or satellite images | [59] |
| | | | Dune Width | | [59] |
| | | | Vegetation cover | | [59] |
| | | | Washovers | | [65] |
| | CF | Forcing variables | Significant Wave height and angle of approach | Bibliographic review or analysis of existing data (e.g., ECMWF or Copernicus program) | [65,67] |
| | | | Tidal Range | | |
| | | Trends | Sea Level Rise | | [13] |
| | | | Storm Surge | | [13] |
| Human pressure | Human Impact Index | Visitors' pressure | Access difficulty | Field work observations | [29] |
| | | | Protected Area Management Category | Bibliographic review or analysis of existing data | [86] |
| | | | Tourism Intensity Rate and Population Density | | Mod. from [87,90] |
| | | for Category III sites | Beach typology | Field work observations | [23] |
| | CF | Trend | Evolution of the number of beds in tourist establishments | Bibliographic review or analysis of existing data | Mod. from [90] |

*Total Sensitivity Index* (spans "Natural processes" and "Human pressure" in the Sensitivity Indexes column)

* to be carried out during summer period.

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
