# Peer review of "An Innovative Approach to Determine Coastal Scenic Beauty and Sensitivity in a Scenario of Increasing Human Pressure and Natural Impacts due to Climate Change"

_water, doi:10.3390/w13010049_

Round 1
Reviewer 1 Report
The article is well written and easy to read. It addresses a significant topic, at the intersection between tourism and climate change studies. It is based on a solid empirical material (29 beaches have been analysed in depth).
The methodology is based on an index assessment approach that considers both natural and human related factors. My comments are the following.
The CSES is based on weights and a grading matrix in order to ponder for human qualitative judgement. Table 2 and Table 6 do not include weights. The authors hence consider all the factors of equal importance/incidence. That should be better justified.
The sensitivity indices are categorised in thee classes, based on arbitrary thresholds (<0.33, 0.33-0.66, > 0.66). The thresholds should also be better justified and possibly tested. I would suggest to make use of the distribution of SI provided in Table 10 to discuss these. For instance the distribution of Total SI varies between 0.32 and 0.75. I would suggest to divide this distribution into three quantiles in order to identify most sensitive and least sensitive beaches according to the index system. It is not possible in my view to state that all class II beaches are of medium sensitivity without further expert evidence.
In other terms, what is proposed here is a classification system, not an evaluation system. The later would require some form of external validation. A classification system is of value in that it helps to orient priorities. In this perspective, it would be helpful to compare CSES evaluations with sensitivity analysis. Is there a relation between both?
Table 4. I can not understand that Dune height can be <1 or absent if we are only assessing beaches with dunes here.
The beach typology parameter appears somehow redundant with previous parameters as for instance access difficulty or human density.
In the conclusion/discussion section, would it be possible to reflect on some automatic procedure for analysing these parameters in the future? It can be understood that the proposed study is based on field visits and visual analysis. Still a number of factors may be collected through statistical databases and/or remote sensing data in a near future. That would greatly enhance the replicability of the approach.
Author Response
Thank you very much for your observations/questions. Please see attached file with detailed answers.

Reviewer 2 Report
Dear authors,
this is an interesting manuscript about a well-known approach used in coastal scenic assessment. This work is an application of the methodology developed by Ergin et al. (2004) and updated by Rangel-Buitrago (2019).
From a formal point of view, I suggest to follow the typical manuscript structure in which we can find:
1-Introduction (state of the art, literature gaps, and study aims)- WITHOUT applied methods. For instance, I think Table 1 and Figure 4 and their explanations should be moved in Methods
2- Study area (separated from/or integrated into Methods) I suggest including information about the geology, climate, tides, winds, etc. of the study area
3- Methods (in the paper is 3. Determination of scenic sites sensitivity processes) WITHOUT the study aims
The introduction is very well-written..however, it seems quite general and long, because it covers different generic aspects (e.g. lines 44-65 coastal economics and tourism; lines 66-95 coastal erosion and conservation; 126-155 scenery and CSES method). From my perspective, it could be better a shorter introduction with some concrete literature evidence about the needs of CSES improvements. In this way, readers could also better understand why you decided to integrate 2 "erosion" and "CC" indexes in a scenery assessment (we know there are a million of erosion evaluation studies that used more than the 2 formulas..).
line 234: Erodability Index: why did you choose to use two rocky parameters? rock erosion is not easy to evaluate. In the study area (Mediterranean basin) subsidence is to be considered. However, A reduction in sediment river input is another (if not the main) cause of erosion in the study area.
line 294-302: Dry beach width as a multiple of the ICZ: Here, I find the problem of this paper that I saw in the use of different methods. You introduced an erosion evaluation that is estimated in long periods (this is good!!), with a CSES method that is very subjected to seasonal changes (in all honesty, I found differences in CSES application in monthly-daily scale, e.g. litter, noise parameters). In the same way for the evaluation of Climate index. Anyway, I think it could be interesting but very challenging to apply your integrated method in the case of local erosion related to individual storms (in this case, the time scale is comparable).
Regarding data collection, I suggest describing the CSES-Erosion-CC-human factors in a summary or a table that is useful to highlight the time-season-coastal sector characteristics and sources (where did you collect all data?) in relation to each data type. How can other scientists replicate your approach?
Washovers (%) you indicated percentages up to ≥50% of washovers. These percentages are quite high and depend on the coastal sector length and the time scale applied to CC. Are these % significative for the Mediterranean case study or for specific beach features? can we apply them at all beach types and CC scales?
Please add the CSES category near D value for each beach of table 10.
Minor typos in the upload file

Author Response

(The authors gave the same response as above.)

Reviewer 3 Report
The paper presents the results of a study focuses on the coastal scenic assessment of 29 coastal sites located along the Mediterranean coast of Andalusia. The paper is well presented, English is correct, there is a good balance between the various parts of the paper, and the topic is interesting and fits with the scope of the journal.
Few recommendations are made in order to improve the final version:
"Introduction": this section is very well written covering different general issues: coastal economics and tourism, coastal conservation and erosion, coastal scenic beauty assessment. References are relevant, anyhow the authors could consider the opportunity to add the following references (see annotated manuscript):
- Prampolini M., Savini A., Foglini F., Soldati M., 2020. Seven Good Reasons for Integrating Terrestrial and Marine Spatial Datasets in Changing Environments. Water, 12, 2221. doi: 10.3390/w12082221
- Rizzo A., Vandelli V., Buhagiar G., Micallef A.S., Soldati M., 2020. Coastal Vulnerability Assessment Along the North-Eastern Sector of Gozo Island (Malta, Mediterranean Sea). Water, 12, 1405. doi: 10.3390/w12051405
- Selmi L., Coratza P., Gauci R., Soldati M., 2019. Geoheritage as a Tool for Environmental Management: A Case Study in Northern Malta (Central Mediterranean Sea). Resources, 8(4), 168. doi: 10.3390/resources8040168
Chapter "3. Determination of scenic sites sensitivity processes": How did you select the three thresholds (<0.33, 0.33-0.66, >0.66)? Please justified better your choice, this could help the reader.
Further comments in the annotated manuscript.
Information about geology, geomorphology, climatic setting ecc. are missing. The authors could consider the opportunity to add these information in a paragraph within chapter 6. Studies sites

Author Response

(The authors gave the same response as above.)

Round 2
Reviewer 1 Report
You could reflect on the relation between CSES and SI. If you plot both variables of Table 10 (I used your table), you get this graph (attached). You can the distinguish beaches with high/low CSES (mean value 0.95) and high/low SI (mean value 0.54). Beaches with both high CSES and high SI may be considered as a priority in terms of policies. This is a first "rough" attempt at crossing both variables. You could do much more obviously...

Reviewer 2 Report
Dear authors,
Your manuscript has been improved, and the second version is better than the first one. You also answered all questions I had (quite all, I still doubt the washover% but it is a detail), so the review is complete from my point of view.
All the best